# Plate Breakage After Mandibular Condylar Fracture Osteosynthesis

**DOI:** 10.3390/jfb16100389

**Published:** 2025-10-16

**Authors:** Marcin Kozakiewicz, Paulina Agier, Paulina Pruszyńska

**Affiliations:** 1Department of Maxillofacial Surgery, Medical University of Lodz, 251 Pomostka Str., 92-213 Lodz, Poland; 2Multispecialty Dental Clinic, 106/116 Kościuszko Av., 90-442 Lodz, Poland

**Keywords:** mandible, condyle, mandible condylar fracture, ORIF, osteosynthesis, fixation, treatment, complications, plate, failure

## Abstract

Despite the significant ongoing development of fixation materials, plate breakages still occur after osteosynthesis of the mandibular condyles. The aim of this study is to demonstrate the complications caused by fixation material breakages in the hope of inspiring the development of new, more durable plates; we analyzed a total of 238 plates used for osteosynthesis in this anatomical region. Cases where compression screws were used as the sole fixation material were excluded. Plate breakage was found in six cases, which accounted for 2.52% of treated individuals. It seems that most plate breakages can be avoided by maintaining effective patient supervision for up to 6 months after surgery. Risk factors for breakage are identified and guidelines for improving the design of future plates are provided. By analyzing some plate design features, we provide some indications for improving their strength and improving their designs for use in this field.

## 1. Introduction

Mandibular fractures are some of the most common facial bone traumas [1], with maxillofacial surgeons playing important roles in providing first aid, definitive treatment, and management of late sequela. These include fractures of the mandibular condyle and the condylar base [2,3], for which treatments have existed for decades [4]; while the approaches to this treatment are relatively straightforward [4,5,6], the fixation materials used have demonstrated various issues.

It appears that the frequency of reoperation after osteosynthesis increases with age, rarely affecting younger people [7]. Cases of reoperation due to plate breakage occur in up to 12% of surgically treated patients [8,9].

Despite significant advances in materials science (such as steel, vitalium, and titanium alloys) [10,11], numerous inventions (such as L-shaped and compression plates) [12,13,14,15,16,17,18], and improvements in knowledge [19,20,21,22] (including biomechanical knowledge [23,24,25,26]), the plates used to fix mandibular condylar bone fragments may still break after osteosynthesis.

Material failure is the most common reason for reoperation after mandible ORIF [27], and may affect up to 80% of second surgeries; the most common factors are poor fixation and reduction. A single straight miniplate should not be used for the fixation of mandibular condylar fractures [28]; it is now well established that the gold standard for osteosynthesis in these cases is the use of two straight plates fixed divergently [29,30]. However, plate breakage is rarely described following osteosynthesis of the mandibular condyle. This applies in particular to damaged dedicated plates, which are generally resistant to breakage [28].

Loss of fixation stability is functionally catastrophic, negating all the advantages of ORIF [31]. Displacement, or worse, dislocation of the proximal fragment lead to malocclusions, such as anterior biting and affecting mastication on the ipsilateral side. Less commonly in this anatomical region, a loss of fixation stability can lead to infection [32]; abscess formation in the space under the masseter muscle and in the pterygomandibular space can escalate into life-threatening parapharyngeal conditions.

In this study, we wanted to determine the conditions under which fixation plates break to improve future plate designs for osteosynthesis in the mandibular condyle region. We present plate breakage as a rare complication of mandibular condylar fracture osteosynthesis, as found in the medical records archive of the Depatment of Maxillofacial Surgery of the Medical University of Lodz, Poland.

## 2. Materials and Methods

This study was a retroscpewctive analysis of medical records from 2017 to 2023. This observational study was reported in accordance with the Strengthening the Reporting of Observational Studies in Epidemiology (STROBE) Statement [33].

Clinical retrospective data were extracted from the hospital database following institutional approval. Medical records of patients referred with mandibular condyle fractures between 2017 and 2023 were identified using ICD codes. The extracted data were subsequently processed, CT scans were analyzed, and all data were then anonymized.

The selection of clinical material was determined by inclusion and exclusion criteria. The inclusion criteria were patients presenting with mandibular condyle fracture treated using plate osteosynthesis or titanium alloy plates. The exclusion criteria were the sole use of long screw fixation, lack of documentation, or no follow-up.

The follow-up period was 24 months after surgery, during which time the patients were examined regularly. Routine examinations were scheduled eight times: immediately after surgery (00M) and at 1, 2, 3, 4, 5, 6, and 24 months post-operatively (01M, 02M, 03M, 04M, 05M, 06M, 24M). At 00M, 06M, and 24M, the follow-up protocol included computed tomography (CT) examinations. Patients who developed complications received individualized treatment and were managed according to tailored follow-up plans.

Using the department’s medical records, we identified 238 cases of condylar fracture treatment that met the inclusion criteria. This group consisted of 57 females and 181 males, with an average age of 45.9 ± 20.3 and 36.5 ± 13.2, respectively (38.8 ± 15.6 years for the entire group). Overall, 69 patients were rural residents and 169 were urban residents. Patients who consumed alcohol prior to injury accounted for 45% of cases. Osteosynthesis was most frequently performed in August (37 procedures) and least frequently in February (4 procedures). Fifteen patients were referred from other maxillofacial surgery departments, while the remaining 223 were diagnosed and treated at the same medical center. All fixing materials were manufactured by ChM (ACP by ChM, Białystok, Poland, www.chm.eu; access date 3 September 2025).

This material was examined for complications in the form of osteosynthesis plate breakage. A typical example of plate breakage is presented below in Figure 1.

The following variables were examined: age, gender, place of residence, cause of injury, alcohol consumption, body mass index, number of co-morbidities, diagnosis, associated mandibular fractures, time interval between injury and surgery, surgical approach, type of osteosynthesis plate used, duration of surgery, facial muscle function [34], wound healing, Helkimo index [35], plate fractures within 24 months after osteosynthesis, and reoperations.

Statistical analysis was performed in Statgraphics Centurion 18 (Statgraphics Technologies Inc., The Plains City, Warrenton, VA, USA; www.statgraphics.com, accessed on 3 September 2025) and consisted of a normality check. The effect of plate breakage on quantitative variables was assessed using one-way analysis of variance (for normal variable distribution) or the Kruskal–Wallis test, while the relationship between breakage and qualitative features was assessed using the χ^2^ independence test. In addition, the Mechanical Excellence Factor (MEF) was calculated and, on this basis, the simple regression of the theoretical load capacity was calculated [36]. The assumptions of the model were as follows: the raw material is titanium alloy 23, the plate thickness is 1 mm, the plate is cut with a laser, and the plate is fixed exclusively with 2.0 system screws with a length of 6 mm. In this way, we attempted to compare 52 plate models from the literature. A *p*-value of less than 0.05 was considered statistically significant.

## 3. Results

In the study group of 238 individuals, plate breakage was observed in six cases, which accounted for 2.52% of patients. In four of these patients, the reduction in the fragments was correct, while in the other two cases a wide fracture gap was observed, i.e., the reduction was incorrect (Figure 2), and appears to have been the cause of breakage in these two cases. In the remaining four cases, one plate breakage was caused by an epileptic seizure, while the other three may have been caused by chewing of overly hard foods too early. The data are presented in Table 1 below. The following factors were not associated with subsequent plate breakage: patient age at the time of injury, place of residence, cause of injury, influence of intoxicating substances at the time of the initial injury, body mass index, number of co-morbidities, level of condylar fracture, multiple condylar fractures, waiting time for treatment, surgical approach, and selection of plates for osteosynthesis.

Breaks in the fixing material were found only in single fractures of the condyle, but they were often accompanied by additional fractures in the mandible. In five cases, the plate fractures were observed up to six months after surgery (1–6 months post-op). In the sixth case (caused by an epileptic seizure), the accident occurred 11 months after osteosynthesis. It should be emphasized that no fractures of a single straight plate, two straight plates positioned divergently, or an XCP were observed (no statistical significance). It is also worth noting that broken plates occurred twice as often in females (*p* < 0.05). Plate breakages caused a deterioration in the functional result (*p* < 0.05), as expressed by the Helkimo index in the 6-month examination (i.e., shortly after secondary surgery due to plate breakage). Secondary surgery was therefore necessary due to the breakage of the fixation material (*p* < 0.001). 

Plate breakage is not related to facial nerve dysfunction. All patients showed normal face movements at the 24-month follow-up (all patients scored 2 on the House-Brackmann scale in the 6-month examination). A salivary fistula formed in one case of breakage.

Logistic regression analysis was performed and the relationship between variables and plate breakage was analyzed. The following variables were studied: age, place of residence, gender, BMI, co-morbidities, intoxicant use at the time of injury, cause of fracture, diagnosis of condyle injury, number of condyle fractures, associated mandibular fractures, delay in surgery, surgical approach, development of salivary fistula, reoperation, Helkimo index 6 months after surgery, and House-Brackmann score 24 months after surgery.

The following factors were found to be independent risk factors for plate breakage: total number of co-existing mandibular fractures; persistent dysfunction, expressed as the Helkimo index score 6 months after surgery; and reoperation (Table 2).

Of the presented cases of plate breakage, reosteosynthesis was performed in four patients. In one of the remaining cases, union and a bone shape close to the anatomical form of the condylar process was observed after removing the plate remnants (this was left as it was), and in the other, it was necessary to restore the height of the mandibular neck with a autogenic bone graft (Figure 3 and Figure 4).

The course of the above procedure is shown in Figure 4.

The force with which the 3AXP blocks the displacement of fixed bone fragments by more than 1 mm was extrapolated. For a calculated MEF = 28.775, this force reaches 19.391 N. The result of the regression analysis is presented in Figure 5. For a complete presentation of the data, the statistical significance for the Square X model (*p* < 0.0001), the correlation coefficient (CC = 0.83), and the degree of fit of the model to the experimental data (R^2^ = 68%) are given. The regression equation is given in the header of the graph below.

## 4. Discussion

The reported frequency of plate breakage in the literature ranges from 0% to 17%, with a tendency for this value to decrease as the number of patients studied increases [40,41,42,43]. Breaks in plates occur at their weakest point, i.e., the hole or bend where the plate was fitted to the bone surface. In one case, a break at the anterior arm of the plate was described, but this appeared to be a breakage passing through the hole at the point where the lambda plate separates at the anterior and posterior arms [42].

Combined plate breakages and screw loosening are often described as cases of “hardware failure” [42]. From a clinical point of view, this is very appropriate, but to assess the clinical strength of the plate alone, it is worth focusing only on those cases where the screws hold the plate well and the plate itself breaks. This may demonstrate methods for improvinh plate design in the future.

In most cases, the complication depends on the surgeon. In the cases analyzed in this study, one out of six plate breakages was the result of an epileptic seizure in a case of drug resistance. Two further cases resulted from the insufficient reduction of the bone fragments. The last three were most likely the result of insufficient patient supervision during the six months following the completion of surgical treatment. Therefore, it appears necessary to consistently ensure the closest possible open reduction of fractures and provide systematic outpatient care to detect bruxism, non-compliance with a liquid diet, dietary deficiencies, etc.

Plate fractures are more frequently observed in females, although the majority of trauma patients are male—a similar situation to cases of facial nerve dysfunction after mandibular condyle osteosynthesis [44,45]. Females suffer more and for longer. It is difficult to explain this unequivocally; it may be that the female skeleton is more delicate and more prone to osteoporosis [46,47,48,49]. This can be partially explained by concurrent bone formation on the outer (periosteal) cortical bone surface during aging, which partly compensates for bone loss and is more pronounced in men than in women, meaning that male internal bone loss is better offset [48]. More women than men may sustain fractures because their smaller skeleton incurs greater architectural damage and adapts less effectively through periosteal bone formation.

Infections and salivary fistulas may also accompany plate breakages. Hammer et al. [43] describe the incidence of many fistulas, in contrast to the clinical material presented in this study. This is probably due to antibiotic prophylaxis regimens [50], changes in operating room standards, and a significant decrease in the number of plate breakage cases.

It is important to ensure that the plates fit the bone surface well [51], with the best option being plates that are individually manufactured using metal powder fusion technologies. However, this method remains time-consuming and expensive. Plates cut from rolled sheet metal can also be fitted well. The aim here should be to achieve a fit that leads to load-sharing osteosynthesis [39].

The identified independent risk factors for plate breakage can also be used to describe the difficulties in treating multiple and complicated fractures. If the mandibular body is also fractured, this significantly destabilizes the stomatognathic system, and the body’s osteosynthesis does not always eliminate torsional stresses. Multiple fractures of the mandible and condyles also contribute to long-term abnormal conditions in the masticatory muscles [52,53]. This describes another risk factor for plate breakage: an elevated Helkimo Index value 6 months after surgery. Reoperations are performed when the fracture proves to be so difficult that, after radiological verification, the fixation needs to be corrected [42]. Corrective re-opening is required in cases of early displacement of fixed fragments, renewal dislocation, misalignment, plate malposition, or collision of the implant material with the foramen or mandibular canal [54,55].

The treatment used after a plate break depends on the condition of the bone fragments. The surgeon is faced with either malunion or pseudoarthrosis [56]. In both cases, treatment is more difficult than primary osteosynthesis. The former situation would appear to be better because the bone can undergo osteotomy, reduction, and reosteosynthesis. However, the results of condylar osteotomy [57] in reoperation are uncertain due to the ischemia of bone fragments and a tendency for bone loss. It is also possible to leave the fragments consolidated in a non-anatomical position, ensuring that they do not cause functional impairment (i.e., a minor displacement or a displacement that can be corrected by prosthetic or orthodontic means) [58,59]. The last option is to leave the fragments in an improperly consolidated position and plan orthognathic treatment [60], with the prerequisite of good temporomandibular joint function. In the case of pseudoarthrosis, the first alternative to consider is bone fragment revision, bone grafting to the defect site, and reosteosynthesis (Figure 3). Total alloplastic joint replacement should also be considered as a second treatment alternative [61]. The higher the pseudoarthrosis is in the condylar process, the more advisable it is to use a joint replacement [62,63].

The first potential preventative measure against plate breakage is structural modification [64]. Plates thicker than the standard 1.0 mm, e.g., 1.2 or 1.3 mm, are dedicated to the mandibular condyle and are manufactured by companies such as ChM or Medartis. Moreover, new materials, such as zirconium dioxide, could also be considered for use [65,66]. Regarding design modifications, it is clear that the use of plates with bridges, but no holes, protects against breakage. It is also known that plates with round holes fix more rigidly than those with oval holes [67]. These are just two examples, but general solutions have been suggested by Mechanical Excellence Factor (MEF) analysis [36].

Plate failure incidents are thought to occur as a result of insufficient reduction or bruxism [68]. Overall, our observations, as presented in the Results Section above, confirm this. Such accidents require reoperation in cases of bone resorption, existing drilled holes, and tissue scarring [69]. Primary surgery in the condylar region requires a good knowledge of anatomy and training in a large maxillofacial center [68]. It is advisable to avoid reoperation due to plate fractures. Open reduction, which is technically challenging and described as “fraught with anatomical hazards” [70], requires an experienced surgeon but has clear advantages regarding post-operative function and anatomy. Over time, these advantages are becoming more widely accepted, and the risk of surgical complications is considered acceptable [44,71]. The use of IMF should be considered to ensure anatomical stabilization of the fracture [68]; however, this results in the loss of ORIF’s significant advantage. It is not necessary to offer it to everyone. Should intermaxillary fixation (IMF) be proposed for patients who the surgeon expects will not cooperate, as they may remove the elastics? Should use wire ligatures be used instead? This, in turn, is not a safe solution; in the event of obstruction, for example, there is no way to quickly unblock the upper airways. Therefore, it seems that the only option is to increase the effectiveness of osteosynthesis, and as such, MEF analysis can be helpful in determining whether a material is sufficient.

The MEF consists of two components that have opposite effects on plate performance [36]:Improvement Component = 0.924362∙Total Fixing Screw Number + 0.708092∙Number of Screws in Condyle + 0.804335∙Height + 0.802964∙Width + 0.752599∙Plate Surface Area + 0.189877% Round Holes − 0.0300946∙Number of Oval Holes in Plate − 0.11967∙Oval Holes Share(1)Deteriorating Component = −0.108505∙Total Fixing Screw Number − 0.109∙Number of Screws in Condyle − 0.0375889∙Height − 0.0761193∙Width − 0.127803∙Plate Surface Area − 0.940781% Round Holes + 0.97506∙Number of Oval Holes in Plate + 0.980953∙Oval Holes Share(2)

The Improvement Component (1) is mainly influenced by (in order of significance) the number of fixation screws, plate height, plate width, plate surface area, and the number of fixation screws in the proximal fragment. Slightly less important, but still positive, is the number of round holes in the plate. These are characteristics that describe rigid plates, promising successful osteosynthesis. Of course, this depends on both the fixation material and the human factor. Therefore, the condylar region requires a traumatologist with a steep learning curve, knowledgeable in surgical anatomy, able to cope with unexpected twists and turns in the operating field, and able to effectively manage complications. The other component of the MEF describes the elements of plate design that reduce the stability of osteosynthesis (Deteriorating Component). These include (2) the number of oval holes among all screw holes and the number of oval holes in the plate. Therefore, the presence of oval holes is an obvious factor that impairs plate mechanics; the lowest score in condylar base osteosynthesis are obtained by 4-hole plate with only oval holes, in comparison with the 51 plates analyzed (Figure 5).

The stiffness of any osteosynthetic plate should be double or even triple that of the mandible in the fractured region to promote physiological bone growth and healing [72]. About 20 years ago, this recommendation was implemented by modifying a straight 6-hole plate with a bridge, which works very well in condylar fixations [73]. Another standard that has been established is plate thickness, which must be at least 1 mm [74]. When verifying future plate designs for the fixation of mandibular condylar process fractures, the MEF is worth using. It is a combined measure of eight design features (total number of fixing screws, number of screws in the condylar part, height of the plate, width of the plate, plate surface area, number of round holes in the plate, number of oval holes in the plate, and the number of oval holes out of the total number of holes in the plate). The previous Plate Design Factor [38] used only four plate design features and does not evaluate the effect of oval holes on plate fixation rigidity, which has a negative impact on osteosynthesis [67]. The MEF values for a given plate design are strongly related to the measured average value of F_max/dL_ [36], making it a good tool for numerically describing the mechanical quality of plate designs. For example, the MEF for a three-axis plate is 28.8; compare this to the scores of the ACP (ACP-T version) (22.1), the XCP (XCP Universal 3 + 5-hole version) (21.9), and the large delta TCP (trapezoid pre-shaped 9-hole version, ref. M2-4860) (21.2). A higher value represents the greater mechanical quality of the design. For a better understanding of the significance of MEF, it is worth mentioning that for the Gold Standard, i.e., a two-plate fixation with straight plates, MEF = 17.8. For small 4-hole delta, square, or rhombus plates, the MEF takes values between 8.9 and 9.4. Therefore, it is possible to design clinically superior solutions in a new design, i.e., one that is more resistant to breakage. This shows that it is still possible to construct rigid plates from currently available raw materials. Furthermore, other possibilities include thickening the plate; using grade 2 titanium; avoiding cutting at elevated temperatures, e.g., using a waterjet; using longer screws, e.g., 8 mm, etc.

This indicates that success should be expected when the plate has large dimensions, more than six holes for screws, a transverse arm connector, and short connectors between holes. These are further guidelines for doctors when choosing plates for their patients. It is also worth monitoring patients after surgery—this depends on the medical team. Other factors that can protect against plate breakage and depend on the surgeon include performing a fixation along the ideal osteosynthesis line [24]; using the full available bone thickness in the condylar process [3]; and performing load-sharing osteosynthesis as opposed to load-bearing osteosynthesis [39] with perfect reduction of the fracture [47,74]. This last piece of advice is not applicable in comminuted fractures and is difficult to apply in old fractures.

The last thing to know is that any plate can be damaged if the patient is not effectively monitored for 6 months after osteosynthesis to avoid bruxism [75,76] and ensure a soft diet is maintained [77,78,79,80,81].

The limitations of this study are its retrospsective protocol, single-center design, and limited number of cases. Future studies on eliminating plate breakages may focus on materials science, plate design, or pharmacological stimulation, and may benefit from a multi-center design.

## 5. Conclusions

Plate breakages after osteosynthesis are relatively rare complications, but are worth bearing in mind, especially during the reduction in bone fragments and during follow-up visits to the outpatient clinic in the six months following surgery.

To reduce the risk of plate breakage in future fixation materials, it is worth considering more robust plate designs. Surgeons should use the most favorable bone conditions to select the plate fixation site, e.g., ideal osteosynthesis lines and areas of bone thickening where longer screws can be inserted.

## Figures and Tables

**Figure 1 jfb-16-00389-f001:**
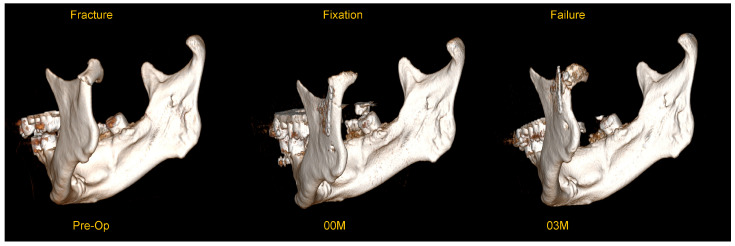
An example of plate breakage after correct osteosynthesis using a dedicated plate for a low-neck fracture of the mandibular condyle. These are three-dimensional reconstructions from a CT scan (RadiAnt software v.2021.1 (64-bit), www.radiantviewer.com/en; access date 3 September 2025). The preoperative image shows a fracture with significant anteromedial displacement. Image 00M (immediately post-operative) shows the immediate result of surgical treatment (open rigid internal fixation)—please note the significant pathological degenerative changes in the joint surface. Image 03M (three months post-operative) shows a breakage of the plate in the upper part which was detected 3 months postoperatively—the break line passes through the upper group of holes in the plate (the fracture most often passes through the holes and not the arms of the plate).

**Figure 2 jfb-16-00389-f002:**
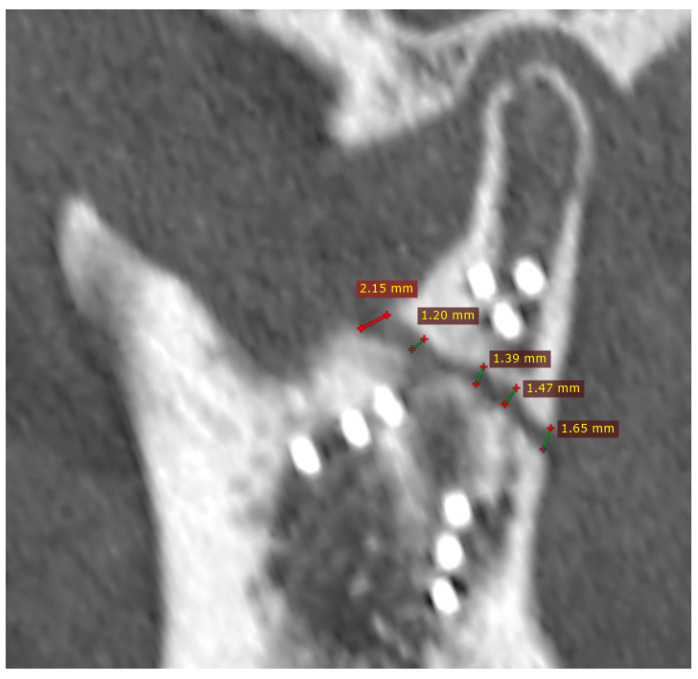
Fixation with incorrect reduction (CT scan immediately post-operative). Despite fixation with nine screws and the proper head position in the glenoid fossa, load-bearing osteosynthesis was created (this probably caused the plate to break when the patient began masticating hard foods in fourth month post-op). This is the result of a wide fracture gap. The distance between the bone fragments ranges from 1.2 mm to as much as 2.15 mm. It is believed that during fixation of the condylar process after a recent fracture, the gap should be reduced to less than half a millimeter, and ideally to a hairline width. This should result in load-sharing osteosynthesis, which is much more reliable than load-bearing osteosynthesis.

**Figure 3 jfb-16-00389-f003:**
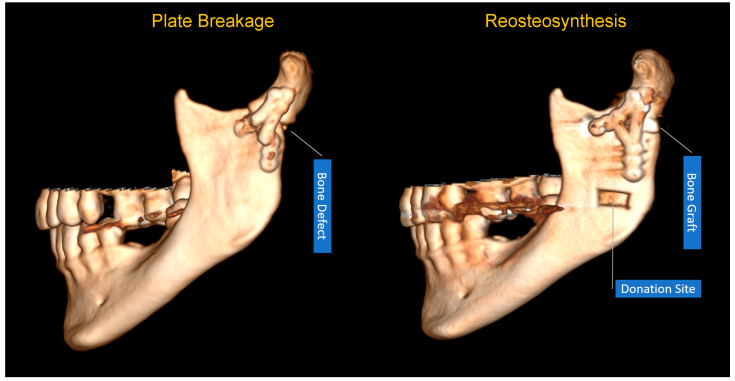
Repair after plate breakage and bone resorption in mandibular neck. In this case, an autogenous bone graft was harvested from a single approach to avoid increasing morbidity (retromandibular approach). Mandibular bone was interpositioned between the base and neck to restore the lost height of the mandibular ramus.

**Figure 4 jfb-16-00389-f004:**
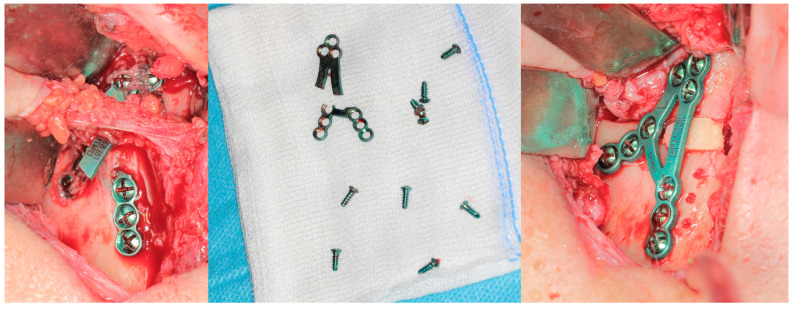
Intraoperative view of a broken plate (**left**). During reoperation, all fixation material (**center**) should be removed and replaced with new material. The anatomical height of the mandibular neck was reconstructed with an autogenous bone graft (**right**). Re-entry is always challenging because the surgeon is required to dissect within the scar, which increases the risk to the facial nerve. Note: no screw is broken.

**Figure 5 jfb-16-00389-f005:**
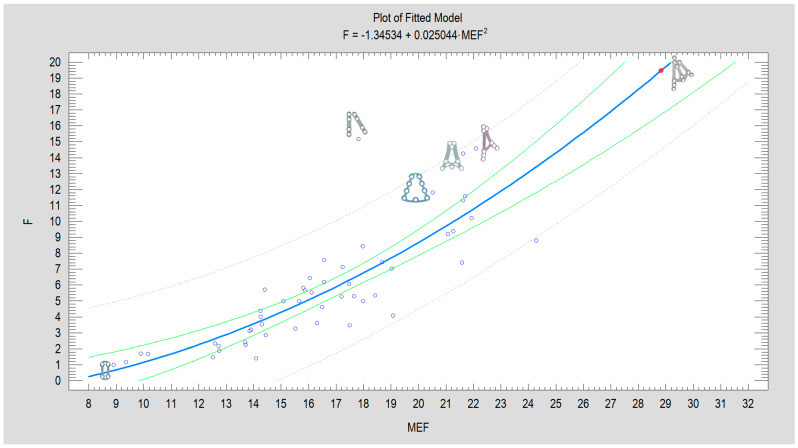
Model of the relationship between the force F (given in N) required for 1 mm displacement of fixed fragments (experimental data) and design advantages expressed as the Mechanical Excellence Factor (MEF; number without a unit). Each blue circle on the graph corresponds to one of the currently known shapes of plates used in mandibular condyle osteosynthesis (bibliographic data [36]). One new plate design, the three-axis plate [39], has been added and marked with a red dot on the plot. Examples of plate shapes illustrating high construction quality (five plates) and low construction values (one plate) have been added next to the circles corresponding to these designs. The blue bold line corresponds to the plot of the calculated regression equation, the green line represents the confidence limit, and the dashed gray line is the prediction limit in this model.

**Table 1 jfb-16-00389-t001:** Clinical data collected for the purpose of analyzing observed breakage of osteosynthesis plates in condylar processes of the mandible.

Variable	Stable Osteosynthesis	Plate Breakage	Significance
Age [years]	38.55 ± 15.59	46.50 ± 17.10	*p* = 0.207 ^a^
Gender	Female/Male = 53:179	Female/Male = 4:2	*p* = 0.046 ^b^
Place of Residence	Rural/Urban = 68:164	Rural/Urban = 1:5	*p* = 0.827 ^b^
Primary Injury Reason *	Assault: 104 Fall: 71 Sport: 5 Vehicle: 46 Workplace: 6	Assault: 1 Fall: 4 Sport: 0 Vehicle: 1 Workplace: 0	*p* = 0.437 ^b^
Intoxicant Use During Injury	No/Yes = 120:112	No/Yes = 5:1	*p* = 0.264 ^b^
BMI [kg/m^2^]	23.19 ± 4.39	24.73 ± 7.23	*p* = 0.689 ^a^
Co-morbidity [n]	0.4 ± 0.8	0.7 ± 0.5	*p* = 0.115 ^a^
Fracture Diagnosis	CHF type A: 1 CHF type B: 7 High-Neck: 4 Low-Neck: 33 Base: 187	CHF type A: 0 CHF type B: 0 High-Neck: 0 Low-Neck: 2 Base: 4	*p* = 0.753 ^b^
Condylar Fracture	Single/Bilateral = 181:51	Single/Bilateral = 6:0	*p* = 0.285 ^b^
Associated Mandible Injury	2.0 ± 0.7	1.3 ± 0.5	*p* = 0.024 ^a^
Delay of Surgery [days]	8.7 ± 8.5	6.2 ± 4.6	*p* = 0.416 ^a^
Surgical Approach	Auricular: 1 Ext. Preauricular: 49 Preauricular: 36 Ext. Retromandibular: 35 Retromandibular: 86 Periangular: 8 Intraoral: 17	Auricular: 0 Ext. Preauricular: 0 Preauricular: 3 Ext. Retromandibular: 2 Retromandibular: 0 Periangular: 0 Intraoral: 1	*p* = 0.129 ^b^
Fixing Material	1 Straight Plate: 4 2 Straight Plates: 73 3 Straight Plates: 5 ACP: 125 XCP: 20	1 Straight Plate: 0 2 Straight Plates: 0 3 Straight Plates: 1 ACP: 5 XCP: 0	*p* = 0.405 ^b^
Duration of Surgery [minutes]	174 ± 78	158 ± 79	*p* = 0.595 ^a^
House Brackmann Scale 06M	1.5 ± 1.0.	2.0 ± 0.0	*p* = 0.266 ^a^
House Brackmann Scale 24M	1.0 ± 0.1	1.0 ± 0.0	*p* = 0.811 ^a^
Salivary Fistula	No/Yes = 214:18	No/Yes = 5:1	*p* = 0.974 ^b^
Helkimo Index 06M	0.56 ± 0.85	1.5 ± 1.22	*p* = 0.030 ^a^
Reoperation	No/Yes = 225:13	No/Yes = 0:6	*p* = 0.001 ^b^

* The secondary injury was epileptic seizure in one case of breakage; CHF—condylar head fracture; Ext.—extended; ACP—“A”-shaped condylar plate [37]; XCP—“X”-shaped condylar plate, presented as Plate 19 in [38]; 06M—data collected 6 months post-operatively; 24M—data collected 24 months post-operatively. ^a^—ANOVA used; ^b^—χ^2^ independence test.

**Table 2 jfb-16-00389-t002:** Independent risk factors for osteosynthesis plate breakage in mandibular condylar region.

Factor	χ^2^	Estimated Odds Ratio	Lower Limit	Upper Limit	*p*-Value
Associated Mandible Injury	6.921	12.765	0.759	214.6	0.0085
Helkimo Index 06M	6.749	0.1974	0.046	0.846	0.0094
Reoperation	43.135	1499.7	34.06	6,6031	0.0001

06M—value recorded during examination 6 months post-operation.

## Data Availability

The original contributions presented in the study are included in the article, further inquiries can be directed to the corresponding author.

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
