# Peer review of "Plate Breakage After Mandibular Condylar Fracture Osteosynthesis"

_jfb, 2025, doi:10.3390/jfb16100389_

Round 1

Reviewer 1 Report

Comments and Suggestions for Authors

This is an interesting study into the complex complication of condylar fracture surgery. The combination of clinical data with biomechanical principles (MEF) is particularly important and beneficial. However, the manuscript requires major revisions to address multiple issues.

After important revisions , this manuscript could be a strong contribution.

  • The introduction is quite brief with not enough background discussion and gap in the literature identification.
  • The aim of the study needs to be reformulated for grammatical and scientific writing. (it should be in past tense, and remove redundancy in the writing of the aim )
  • Type of the study should be explicitly stated in the aim and methodology
  • The single-center, retrospective design is a limitation. Please explicitly state this in the discussion section.
  • Clarify the follow-up protocol. Was it standardized for all 238 patients or was it unclear in the documentation? Given the data collected it seems that followup protocols were carried for the patients
  • In Table 1: The p-values for "Associated Mandibule Injury" (p=0.024) and "Helkimo Index 06M" (p=0.030) are significant, but they are not identified as independent risk factors in the following logistic regression in Table 2, which instead finds "Associated Mandible Injury" significant. Explain this discrepancy . Was it due to collinearity with other variables in the multivariate model? Clarify the relationship between these findings.
  • For the logistic regression (Table 2), please provide 95% Confidence Intervals for the Odds Ratios to give a better sense of the estimate's precision
  • Explicitely state MEF analysis was based on which previous biomechanical studies
  • The statement on Page 9: "The stiffness of any osteosynthetic plate should double or even triple the stiffness of the mandible..." is a strong biomechanical claim. Please provide a reference to support this specific recommendation, and consider tempering such statements
  • The discussion on Page 8 (delicate skeleton, osteoporosis) is speculative. Please temper such statements
  • The conclusion that "most of observed here complication is possible to avoid by surgeon" is too direct. Kindly tone down
  • Page 3, Figure 1 Caption: "Image 00M" and "Image 03M" – please define these abbreviations in the caption (e.g., 00M: immediate postoperative; 03M: 3 months postoperative)
  • The statement"Helkimo index was 0 in two cases, 2 in three cases and 3 in one cases"  this sentence is redundant, as the data is already in Table 1, so you can remove it
  • Figures and graphs are good
Comments on the Quality of English Language

some typos and grammatical inconsistencies throughout the manuscript

Author Response

Please, find attached corrected manuscript (my reply has been written in blue color text).

Reviewer 2 Report

Comments and Suggestions for Authors

A research article more oriented to surgical technique and less to material factor.

Introduction:

1)Among fixation plates materials zirconia materials are missing. It would be helpful to include it in the list of them making a comment if they are still an alternative choice.

2)In general introduction is relatively short, one or two paragraphs analysing the existing materials for plates and an explanation why these materials are fragile. Limitations in dimensions, material fatigue, phase transformation, crack propagation, mechanical properties etc could be shortly analyzed.

3) Discussion: The factors surgeon and patient are inserted. This should be commented also in introduction and deeper analyzed in discussion part. Moreover, lines 202-211 should be rewritten it is not clear what is stated, based on female complications.

4) Limitations of this research should be added and future research for the elimination of breakage in plates.

5) A table for the classification of plates (positive / negative points, holes, shapes etc) would be helpful for the readers.

Author Response

(The authors gave the same response as above.)

Reviewer 3 Report

Comments and Suggestions for Authors

The current research conducted by Kozakiewicz et al. aimed to present the complication of fixation material breakage, with the hope of prompting the development of new, more durable plates.

This manuscript offers valuable insights with the potential to contribute significantly to clinical practice and ongoing scientific research. The following comments and suggestions are intended to help the authors enhance clarity, improve the structure, and strengthen the overall scientific quality of the manuscript.

  1. Introduction

The introduction provides an overview of the specialized literature on fixation plates used for mandibular condylar bone fragments, and highlights material failure as the most common reason for reoperation after ORIF.

To enhance the introduction, it would be beneficial to include a separate paragraph before presenting the study’s aim, emphasizing the novelty of the research. This section should clearly articulate how the current study differs from previous work, highlighting its unique features and specific contributions to the field. Including such a paragraph will help clarify the study’s relevance and strengthen the overall impact and significance of the manuscript.

Following the statement of the study's aim, it would be beneficial to include a bullet-point list outlining the main objectives of the research. Such a structure would help clarify the study’s investigative directions, making the scientific contribution and applied methodology easier to understand. Additionally, it could improve the readability and coherence of the introduction by providing the reader with a clear overview of the research approach.

  1. Materials and Methods

The Materials and Methods section provides a comprehensive description of the study's ethical considerations, sample selection and eligibility criteria, and statistical analysis.

When the study variables are listed (“The following variables were examined”), it is recommended to replace the term “sex” with “gender.” Using this term ensures a more comprehensive and sensitive approach to participant diversity, in line with current best practices in scientific research.

  1. Results

To ensure a more coherent narrative flow and clear structure in the manuscript, it is recommended to first present a descriptive paragraph that introduces and summarizes the essential information related to the analyzed data or results. This paragraph should provide the reader with clear context and a preliminary understanding of the main points before referring to a specific table or figure. Subsequently, the specific reference to the relevant table or figure will support and illustrate these points, facilitating smoother reading and a better connection between the text and the illustrative material. After this presentation, a clear and natural transition to the next table or figure can be made, thereby maintaining continuity and coherence in the exposition. This approach contributes to the clarity and professionalism of the presentation, preventing abrupt interruptions in the narrative thread and enhancing the informational impact of the manuscript.

Additionally, after each paragraph, it is important to clearly indicate the figure or table being referenced. For example: “In the remaining four cases, one plate breakage was caused by an epileptic seizure, while the other three may have been caused by chewing overly hard foods too early, as presented in Table 1.”

To enhance transparency and facilitate interpretation, it is advisable to specify the exact statistical tests used below each table. This will enable readers to more effectively evaluate the suitability of the analyses and the reliability of the results.

  1. Discussion

The Discussion section is well-organized and thoughtfully developed, offering a clear and balanced interpretation of the results in relation to existing literature. The arguments are supported by appropriate references, and the study’s strengths and limitations are openly acknowledged, enhancing the manuscript’s overall credibility.

At line 235, it is recommended to remove the word “see” and retain only the reference (Figure 3).

To further enhance the quality and impact of the manuscript, it is recommended to include two new subsections titled “Limitations of the Study” and “Recommendations for Future Research.”

The Limitations subsection would provide a transparent discussion of the study’s constraints, helping readers to better understand the context and boundaries of the findings.

The Recommendations subsection could offer clear, focused guidance on potential directions for subsequent research, grounded in the gaps and challenges identified within the current study.

Incorporating these sections would not only add scientific rigor and value to the manuscript but also serve as a useful framework for the academic community, facilitating the ongoing advancement of knowledge in this field.

  1. Conclusions

The Conclusions section is well-crafted, providing a clear and concise summary of the main findings of the study.

Author Response

(The authors gave the same response as above.)

Round 2

Reviewer 1 Report

Comments and Suggestions for Authors

Reviewer comments were addressed adequately. The reviewer would like to commend the authors on the tremendous improvement of the manuscript.

Reviewer 2 Report

Comments and Suggestions for Authors

Much improved manuscript. English editing transformed the whole text.

All reviewers comments were taken under serious consideration. Discussion part was completed transformed, the authors analyzed much deeper the causes of this rare but very serious complication of osteosynthesis plate breakage.